# ADAPTER-RL: ADAPTATION OF ANY AGENT USING REINFORCEMENT LEARNING

## ABSTRACT

Deep Reinforcement Learning (DRL) agents frequently face challenges in adapting to tasks outside their training distribution, including issues with over-fitting, catastrophic forgetting and sample inefficiency. Although the application of adapters has proven effective in supervised learning contexts such as natural language processing and computer vision, their potential within the DRL domain remains largely unexplored. This paper delves into the integration of adapters in reinforcement learning, presenting an innovative adaptation strategy that demonstrates enhanced training efficiency and improvement of the base-agent, experimentally in the nanoRTS environment, a real-time strategy (RTS) game simulation. Our proposed universal approach is not only compatible with pre-trained neural networks but also with rule-based agents, offering a means to integrate human expertise.

## 1 INTRODUCTION

Deep Reinforcement Learning (DRL) agents face challenges when tasked with problems outside of their training distribution, especially those they haven't experienced during training. Packer et al. (2018) point out that current RL algorithms easily overfit to a fixed environment, because they are usually trained on the test set. The sensitivity to environmental changes means DRL models often must be trained from scratch when encountering new tasks. Compounding this, the sample inefficiency inherent to DRL algorithms can leave them ill-equipped for unencountered scenarios at inference, considering their reliance on vast sample sets to grasp basic behaviors. And the dynamic nature of reinforcement learning data adds to the intricacy, as agents refine their strategies, the data they gather evolves, potentially leading to learning complexities or even instability if not judiciously addressed. Equally critical is the balance DRL agents must maintain between exploration and exploitation; leaning too heavily on known tactics can undermine their competence in novel situations. Additionally, catastrophic forgetting presents a significant hurdle, particularly when agents learn tasks in sequence, causing a performance dip in previously learned tasks as they adjust to new ones.

When employing an expert demonstrator, behavior cloning Pomerleau (1988); Bain & Sammut (1995), also referred to as imitation learning, becomes a viable method to train an agent. However, this approach is not without its challenges. Firstly, the success of imitation learning hinges critically on the caliber of the demonstrations. Should the expert exhibit a mistake or suboptimal behavior, the agent is inclined to replicate such discrepancies. Secondly, a distribution mismatch often arises in behavior cloning between the states the expert accesses and the states the agent encounters post-deployment. This mismatch can lead to the agent behaving unpredictably in unfamiliar situations. Complicating matters further, an agent, unlike the expert, is prone to errors. Such mistakes can land the agent in unfamiliar terrain, risking an escalation of errors as the agent strays from the expert's state distribution. Additionally, behavior cloning does not furnish the agent with a feedback mechanism akin to the rewards in reinforcement learning. This absence means that post-training, the agent lacks the innate capacity to identify and amend its erroneous decisions. Lastly, an agent's proficiency is tethered to the expert's capabilities, meaning that the agent's performance will often not surpass that of the expert.

To address these inherent limitations, scholars have pioneered methods fusing imitation learning with reinforcement learning, notably Dataset Aggregation (DAgger) Ross et al. (2011) and Generative Adversarial Imitation Learning (GAIL) Ho & Ermon (2016). DAgger operates by perpetually

interacting with the environment using strategies derived from behavioral cloning to generate fresh data. With this new data, DAgger solicits examples from the expert's strategy, retrains using behavioral cloning on the augmented dataset, and subsequently re-engages with the environment, iterating this process. This method, powered by data augmentation and continuous environment interaction, significantly reduces the instance of unvisited states and, in turn, the error margin. Nevertheless, DAgger demands impeccable expertise quality. GAIL, on the other hand, is underpinned by the Generative Adversarial Networks (GANs) Goodfellow et al. (2014) framework. In this arrangement, the agent, playing the role of the generator, endeavors to produce trajectories that mirror those of the expert. Concurrently, a discriminator works to differentiate between the two. The agent then garners rewards for deceiving the discriminator, enabling it to receive feedback, analogous to rewards in RL, without any explicit reward cues. Yet, GAIL requires meticulous calibration, typical of most GAN-oriented strategies. Its lack of a concrete reward function can render training intricate since the agent's primary objective becomes duping the discriminator rather than honing in on the genuine task at hand. In addition, expert strategies are also used to restore unknown reward functions Ng et al. (2000); Arora & Doshi (2021). Abbeel & Ng (2004) proposed Apprenticeship Learning, whose algorithm terminates in a small number of iterations, and even though it may not be able to completely restore the expert's reward function, the policy output of the algorithm will achieve performance close to that of the expert. Expert-based agent training methods often have difficulty in obtaining agents that are better than experts. When the experts' strategies are not perfect, it is difficult to obtain satisfactory agents. Ye et al. (2020b) uses data deletion and other methods to obtain better agents, but this method is costly in terms of data prepossessing.

In the broad arena of Deep Learning (DL), adapters have gained prominence. An "adapter" denotes a succinct module tailored to fine-tune a pre-trained neural network. Originally proposed for computer vision models Rebuffi et al. (2017) and later extrapolated to NLP Houlsby et al. (2019), subsequent advancements include AdapterFusion Pfeiffer et al. (2021), which proposes amalgamating adapter parameters for multi-task knowledge consolidation; AdapterDrop Rücklé et al. (2021), notable for its innovative pruning mechanism; Compacter Karimi Mahabadi et al. (2021), which refines an adapter structure for superior performance with minimal parameter addition; and various other applications like MAD-X Pfeiffer et al. (2020) for modular knowledge storage, and research into Vision Transformers Marouf et al. (2022) and vision-and-language tasks Sung et al. (2022). A common adaptation method uses a serial structure, that is, inserting a low-rank feedforward neural network into the pre-trained model. Hu et al. (2021) proposed a parallel structure that adds a bypass module to the model. This method will not affect the computational efficiency of the original base large model, and the trained modules can be directly merged into the large model parameters during inference. And this method is also widely used in the field of image generation.

Despite these advancements, the potential of adapters in the domain of reinforcement learning remains largely untapped. Using an adapter to fine-tune the demonstrator may be a feasible method. The adapter allow neural networks to pivot to new tasks without extensively retraining the entire model. The advantages of adapters are manifold. They are known for their parameter efficiency, necessitating only a subset of parameters, which accelerates training, conserves memory, and mitigates overfitting risks, especially with smaller datasets. This efficiency is also conducive for model storage and dissemination. Notably, adapters ensure the preservation of the original model's parameters, addressing the persistent issue of "forgetting" in continuous learning and guaranteeing that foundational knowledge remains untouched. Furthermore, adapters optimize multi-task learning, enabling the simultaneous learning of multiple tasks with minimal parameters. This is a departure from conventional multi-task learning, and while it might slightly limit the mutual advantages drawn from diverse tasks, it reduces task interference.

The weak generalization and catastrophic forgetting of reinforcement learning are difficult to ignore in some complex problems. Even if the current Go AI Silver et al. (2016; 2017); Wu (2019) can easily defeat the top human players under normal circumstances, in some cases it will still make low-level mistakes that are difficult for humans to understand and lose Wang et al. (2022). RTS games are a complex task for reinforcement learning, and contain multiple different maps, which makes it difficult for the agent to perform well on every map, especially maps that are not in its training set. MicroRTS Ontanón (2013) is a simplified RTS game with a long history of running competitions [1] Ontañón et al. (2018). Perhaps due to the requirements of multiple maps, deep

---

[1]https://sites.google.com/site/micrortsaicompetition/competition-results

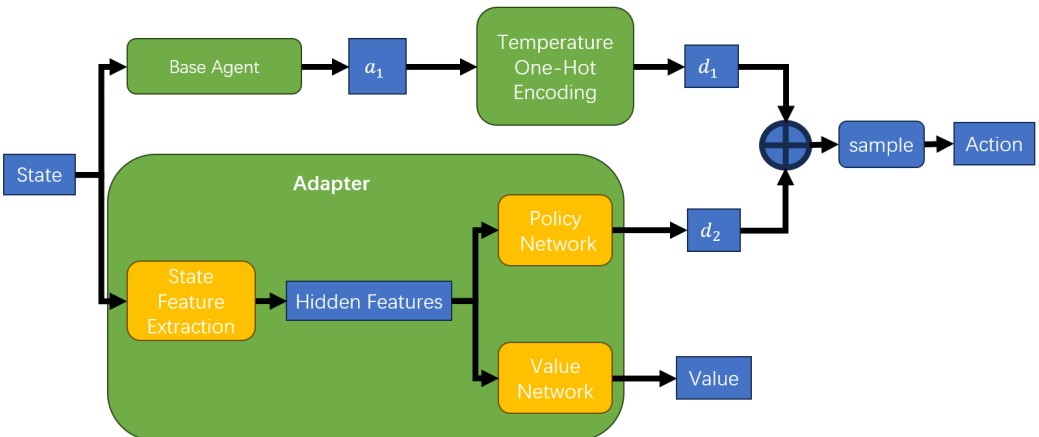

Figure 1: ADAPTER-RL architecture. A base agent receives current state and receives output action $a_1$ which is transformed to action distribution $d_1$ through one-hot encoding. The Adapter is a neural network based on the Actor-Critic framework, where the policy network generates the policy distribution and the value network provides the value estimation of the state. The Adapter receives the state and outputs and adjusted distribution $d_2$. $d_1$ and $d_2$ are added then a sample is taken to determine the final action.

reinforcement learning algorithms have not been used in competitions until this year. This year a bot [2] that used deep reinforcement learning algorithms stood out in the competition, training neural networks individually for nearly every map in the competition, which required a lot of sampling. However, the method did not result in higher scores than the previous 2021 winner. Huang & Ontañón (2021) tested the generalization of the agent trained by the reinforcement learning algorithm in MicroRTS, and its experiments showed the difficulty of generalizing the agent to different maps. We conducted experiments in nanoRTS to test the method under different maps and agents. NanoRTS is a Python-centric version of MicroRTS, a real-time strategy game simulation designed for reinforcement learning research.

Our study proposes a concise and effective adaptation strategy for reinforcement learning. We note that our method can be applied to any agent, including pre-trained neural networks and rule-based agents. This allows human knowledge to be applied to the model, thereby reducing the sampling and training time. Experimentally, we demonstrate our proposed method achieves high training efficiency and stability. The main contributions of this work are:

- We propose Adapater-RL, a novel method that combines reinforcement learning and adaptation to adapt any agent using reinforcement learning.

- Our experimental results demonstrate that the adapter can adapt the base-agent to new tasks more effectively.

- This paper also studies the trade-off of the temperature coefficient in our method.

## 2 METHOD

We propose an adaptation strategy for reinforcement learning. This strategy has similar characteristics to other supervised learning adapter methods: it allows the model to be trained individually for tasks, without the need to train simultaneously on all tasks to avoid catastrophic forgetting. And it only requires a small number of additional parameters to adapt to each new task. Furthermore, it is flexible in that it can be used to fine-tune any agent, not just neural network-based agents.

---

[2]https://github.com/sgoodfriend/rl-algo-impls/blob/main/rl_algo_impls/microrts/technical-description.md

The model we propose is structured around a modular framework as Figure 1 shows, which comprises two branches, with key components consisting of a "base agent" and the "adapter". The base agent's role is foundational. It acts as the primary decision-making entity of the system, providing initial predictions or actions based on its training and inherent capabilities. The base agent is similar to the pre-trained model in other adaptation methods, except that it can be any agent. The adapter acts as a supplementary module to the base agent. Its primary role is to refine and adjust the decisions made by the base agent to ensure they are well-suited to specific tasks. Instead of directly intervening in the internal workings of the base agent, the adapter functions more as a "side branch". Once the base agent delivers its action distribution (a set of potential actions and their corresponding probabilities), the adapter steps in to generate an adjustment distribution. This adjustment distribution essentially represents modifications or fine-tuning to the original action set proposed by the base agent. By combining the base agent's action distribution with the adapter's adjustment distribution, the system can produce a modified action set. This resultant action set is more closely aligned with the requirements of the specific task at hand, ensuring better performance and adaptability.

## 2.1 PROXIMAL POLICY OPTIMIZATION ALGORITHMS

We use proximal policy optimization algorithm (PPO) Schulman et al. (2017) to train the adapter. PPO is a type of Reinforcement Learning algorithm that has been widely adopted because of its effectiveness and stability. Unlike traditional policy gradient methods that adjust the policy in large steps, PPO takes controlled steps to update the policy, ensuring that the new policy is not too different from the old one. In policy optimization, we want to maximize the expected cumulative reward. This is done by adjusting the policy parameters in the direction that increases the likelihood of taking actions that lead to higher returns. However, making large policy updates can lead to sub-optimal policies or even make the training unstable. PPO limits the change in the policy in each update, ensuring the new policy is "proximal" to the old one. This is achieved by adding a constraint to the optimization problem or, equivalently, by adding a penalty to the objective function. Complex problems such as Berner et al. (2019)and Ye et al. (2020a) verify the effectiveness of this method.

Equation 1 is the policy gradient objective of PPO:

$$L_\theta^{CLIP} = \hat{\mathbb{E}}_t \left[ min(\rho_t(\theta)\hat{A}_t, clip(\rho_t(\theta), 1-\epsilon, 1+\epsilon)\hat{A}_t) \right] \tag{1}$$

where $\rho_t(\theta) = \frac{\pi_\theta(a_t|s_t)}{\pi_{\theta_{old}}(a_t|s_t)}$, $\pi$ is a stochastic policy, $s$ is state, $a$ is action, $\theta_{old}$ is the vector of policy parameters before the update, $\hat{A}_t$ is the generalized advantage estimation (GAE) Schulman et al. (2015) of the action at time $t$, $\epsilon$ is a hyperparameter that defines the range in which the policy update is allowed.

At the same time, in our implementation, the value estimator's objective is as Equation 2, where where $V_{\theta_{t-1}}^\pi$ are estimates given by the last value function, and $\hat{A}_t$ is the GAE of the policy

$$L_\theta^{VF} = \hat{\mathbb{E}}_t \left[ (V_{\theta_t}^\pi(s_t) - (V_{\theta_{t-1}}^\pi + \hat{A}_t))^2 \right] \tag{2}$$

## 2.2 TRANSFORMING DETERMINISTIC ACTION TO ACTION DISTRIBUTION

In our proposed structure, the base agent will output a deterministic action for the current state. In order to adjust it with the adjustment distribution output by the adapter, we must convert this deterministic action into a distribution. When converting determined actions of an agent into action distributions for discrete actions, the goal is often to create a soft policy (a distribution over actions) from hard demonstrations (specific actions). This could be useful for training agents in a way that allows for some exploration or smoothing out agents that might have noise.

In our experiments, the environment, nanoRTS, has a discrete action space. There are varied methods to transform deterministic actions into more probabilistic action distributions. One such technique is the One-Hot Encoding with Temperature-Scaled Softmax. In this method, the action is represented as a one-hot encoded vector, which is then passed through a temperature-scaled softmax operation. The temperature parameter is designed to modify the sharpness of the distribution. Another method is Mixing with a Prior, where the deterministic one-hot encoded action is combined with a predetermined distribution, such as a uniform one. A coefficient dictates the balance

between the deterministic agent action and this prior distribution, giving flexibility in determining the resultant mixed distribution. The Additive Noise with Softmax technique is another approach wherein noise is intentionally introduced to the one-hot encoded action. Subsequent to this noise introduction, a softmax operation renders a valid probability distribution across the possible actions. Lastly, in scenarios where multiple agents are present, the behavioral mixture of agents approach, for example Vinyals et al. (2019) samples the final agent from the Nash distribution of the set of agents, can be utilized. Given that different agents, or experts, may recommend varying actions for an identical state, this results in an intrinsic stochastic policy, taking advantage of the diversity in agent decisions. If the state space is continuous, a common approach is to transform the actions into a normal or beta distribution.

We apply one-hot encoding with temperature-scaled softmax. A discrete action space can be represented as a one-hot encoded vector, For instance, if action 2 out of 5 is chosen, its one-hot representation is $[0, 1, 0, 0, 0]$, the scale the one-hot vector to $[0, 1/\tau, 0, 0, 0]$. The higher the temperature coefficient $\tau$, the more spread out the distribution becomes, while a lower temperature coefficient nudges the distribution closer to a deterministic action. The final distribution obtained by mixing the base agent and adapter is shown in Equation 3, where $a_i^{base}$ is the value of the $i$-th action in the base agent action distribution, $a_i^{adj}$ is the value of the $i$-th action in the adjustment distribution.

$$p(a_i) = \frac{exp(a_i^{base}/\tau + a_i^{adj})}{\sum_j exp(a_j^{base}/\tau + a_j^{adj})} \tag{3}$$

If the base-agent outputs continuous actions, take the normal distribution as an example, the temperature coefficient $\tau$ is the standard deviation $\sigma$ in the normal distribution formula, as Equation 4, where $f(a)$ is probability density of action $a$, $a^{base}$ is base-agent action, $a^{adj}$ is adapter agent action.

$$f(a) = \frac{e^{(a^{base}-\mu)^2/(2\sigma)^2}}{\sigma\sqrt{2\pi}} + a^{adj} \tag{4}$$

## 2.3 TRAINING THE ADAPTER

We use PPO to optimize the adapter, which is a actor-critic paradigm Konda & Tsitsiklis (1999). It trains a policy to give action distribution under state and a value function to estimate state.

The algorithm is shown in Algorithm 1. The method starts by obtaining the state representation, $s$, from the environment. Once acquired, the method leverages the base agent to produce either a deterministic action or an action distribution for the given state. In cases where the action is deterministic, it is essential to convert it into a soft action distribution, potentially using previously mentioned techniques such as temperature-softmax.

Next, the method uses the state as an input to the adapter. Ideally, the adapter will output an adjustment distribution over actions. By taking into account the combined action distribution—derived from both the base agent and the adapter's outputs—one can effectively interact with the environment. This interaction will facilitate the collection of trajectories including states, actions, rewards, and subsequent states.

To further refine the process, we compute advantages using the gathered rewards and value estimates. These advantages play a pivotal role as they assist in determining the efficacy of the taken action in relation to the average action for that particular state. Once getting these insights, we proceed to calculate the PPO objective for updates. The primary goal here is to maximize the PPO objective using gradient ascent. Doing so will amend the parameters of the adapter, ensuring its outputs are aligned with more desirable rewards.

Additionally, we note that alongside the policy update, PPO also updates a value network. This network is vital for estimating the anticipated returns. Typically, updates are executed by minimizing the mean squared error, which is determined by measuring the difference between predicted values and actual returns.

---

**Algorithm 1** Adaptation training with PPO

---

**Input:** Iterations $N$, Sample length $T$, Initialized policy parameters $\theta$

**for** iteration=1, 2, . . ., $N$ **do**

    **for** $s_t$ in $s_1, s_2, ..., s_T$ **do**

        Get temperature-scaled one-hot encoded action from base agent

        Get adjustment distribution from adapter

        Get action distribution $p(a_i) = \frac{exp(a_i^1/\tau + a_i^2)}{\sum exp(a_j^1/\tau + a_j^2)}$ and sample to get action $a_t$

        Interact with the environment, get $s_{t+1}, r_t, d_t$

    **end for**

    Get trajectories $(s_1, a_1, r_1, d_1..., s_T, a_T, r_T, d_T)$

    Compute advantage estimates $\hat{A}_1, ..., \hat{A}_T$

    **for** epoch K **do**

        Compute loss $L$, where $\rho_t(\theta) = \frac{\pi_\theta(a_t|s_t)}{\pi_{\theta_{old}}(a_t|s_t)}$

        Optimize adapter with $L$ wrt $\theta$, with minibatch size

    **end for**

    $\theta \leftarrow \theta_{old}$

**end for**

---

## 3 EXPERIMENTS

We conducted experiments in a context defined by expansive state and action spaces coupled with sparse rewards. MicroRTS, as described in Ontanón (2013), is a streamlined version of an RTS game created in Java, which comes with a Python interface named Gym-MicroRTS Huang et al. (2021). We focused our experimental efforts on nanoRTS, a Python-oriented version of MicroRTS. NanoRTS, compared to its predecessors, is more intuitive for Python experts and provides enhanced adaptability for bespoke modifications aligned with research objectives. Being tailored specifically for Python-based reinforcement learning, nanoRTS seamlessly aligns with deep learning techniques [3].

Our foremost aim was to evaluate how effectively our adapter method supports agents in adjusting to different tasks. To offer a clear perspective, we compared the outcomes of our adapter-enhanced method with agents that are solely trained using neural networks on a variety of maps, which symbolize different tasks.

Two foundational agents were employed for the comparison: one rooted in a rule-based AI framework, and the other built upon a neural network architecture. The adapter's architecture consists of a two-layer convolutional neural network (CNN) which feeds into a fully connected multi-layer perceptron (MLP) boasting three layers, each containing 512 units. Proximal Policy Optimization (PPO) was chosen as our training algorithm. We adhered to established best practices for setting hyperparameters: a discount factor ($\gamma$) of 0.99, Generalized Advantage Estimation ($\lambda$ for the GAE) parameter set at 0.95, a PPO clipping coefficient of 0.2, and a value coefficient valued at 1. For optimization, we employed the Adam optimizer with a learning rate of 2.5e-4. To ensure the robustness of our findings and account for potential variance, each experimental setup was executed thrice, with each iteration initiated with a random seed.

### 3.1 ADAPTER WITH RULE-BASED AGENT

In order to examine the effectiveness of our method, we train an agent using an adapter and an agent using only neural networks on different tasks. Initial states of different maps are illustrated in Figure 2. The winning rate against the opponents is used in the game as the main metric and the training curve is also of key importance. At the same time, it is also necessary to refer to whether the trained adapter agent can exceed the base-agent.

In our trials, we utilize a rule-based AI, grounded in the path-finding algorithm, as the foundational agent for our method. Historically, this algorithm showcased impressive results in past MicroRTS

---

[3]Considering the anonymity we will make the code public after the reviewer process

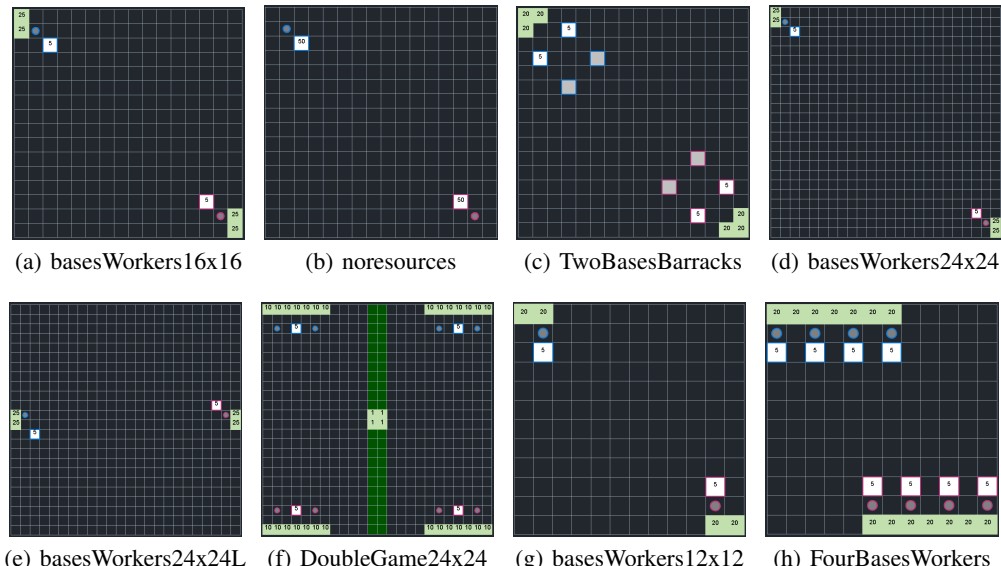

(a) basesWorkers16x16  (b) noresources  (c) TwoBasesBarracks  (d) basesWorkers24x24

(e) basesWorkers24x24L  (f) DoubleGame24x24  (g) basesWorkers12x12  (h) FourBasesWorkers

Figure 2: Initial state of different maps of nanoRTS.

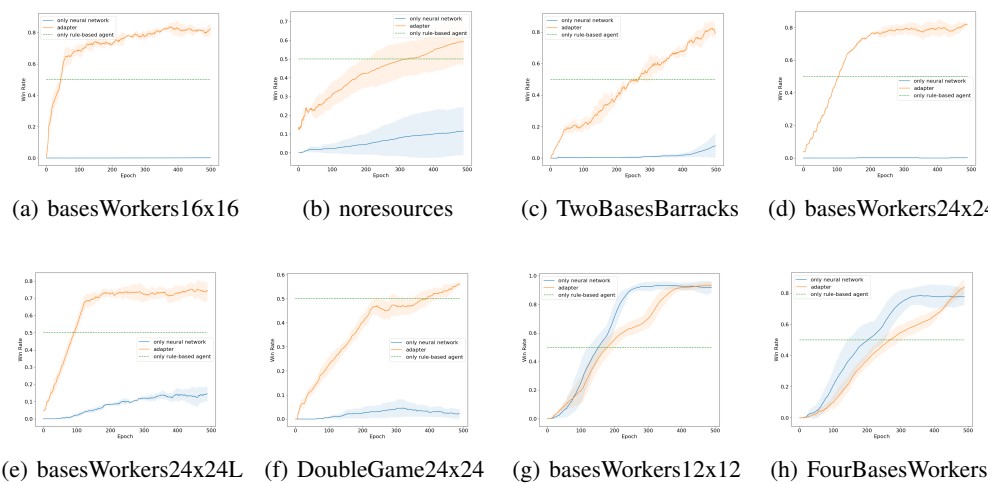

(a) basesWorkers16x16  (b) noresources  (c) TwoBasesBarracks  (d) basesWorkers24x24

(e) basesWorkers24x24L  (f) DoubleGame24x24  (g) basesWorkers12x12  (h) FourBasesWorkers

Figure 3: Training curves with different maps. In each figure, the orange curve is using the adapter, the blue curve is using only the neural network, and the green curve is the winning rate using only the base-agent.

tournaments, clinching victories in both the 2020 and 2021 editions. The agent, in essence, prioritizes targets for each unit under its control, subsequently employing a pathfinding algorithm for unit deployment. Concurrently, this AI was leveraged as the adversary in our tests, resulting in a near-even win rate of approximately 0.5 against the opponent agent. For comparison, we introduced a control group: an agent embedded with a convolutional neural network containing two residual blocks, dependent solely on the neural network. This model underwent training over 500 iterations, with each iteration incorporating 8192 samples.

Figure 3 illustrates the training trajectories across diverse maps for both the rule-based agent and the adapter strategies versus the neural network-centric method. In the majority of our experimental tasks, the adapter drastically hastened the training process and consistently outperformed the base-agent. In contrast, agent training using only neural networks often requires long exploration times until the winning rate starts to increase. However, in scenarios with limited state and action spaces,

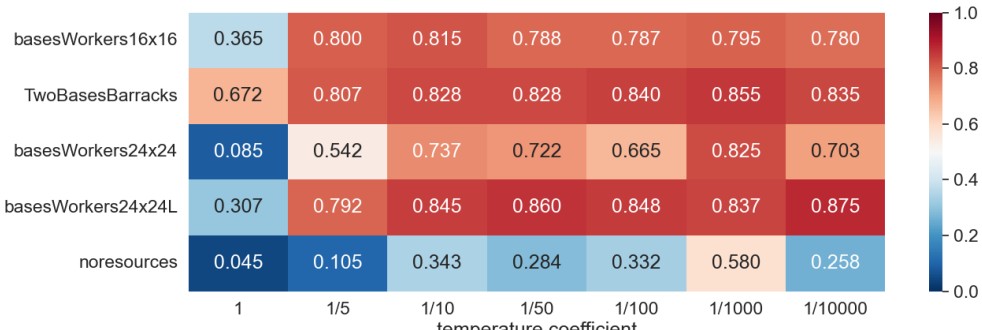

Figure 4: Winning rate after training in different maps with different temperature coefficient

exemplified in Figures 3(g) and 3(h), the adapter's benefits weren't as pronounced. For straightforward tasks, relying solely on neural networks proved sufficient to rapidly develop a high-performing agent, negating the necessity for an adapter. Conversely, in intricate challenges, the utility of the adapter becomes clear.

## 3.2 TEMPERATURE COEFFICIENT TRADE-OFF

The temperature coefficient within our approach regulates the entropy of the action distribution generated by the base-agent. A diminished coefficient drives the action distribution toward a more deterministic result, whereas an augmented coefficient diversifies the distribution. We next explore the impact of temperature coefficients on adapter training in different maps. The experiment settings are the same as the experiment in the previous section.

The result of various tasks are depicted in Figure 4. With heightened temperature coefficients, the distribution derived from the base-agent resembles a uniform distribution. This challenges the distribution's capability to capture the underlying strategy of the base-agent. However, in this case, the results of our experiment are still better than training using only a new initial neural network. This shows to some extent that this method is also effective under a less directive base-agent policy. Consequently, training an adapter under these conditions is analogous to training an agent relying solely on neural networks. Conversely, a reduced temperature coefficient results in the agent's distribution mirroring closely the strategy of the base-agent. This can suppress the agent's explorative tendencies and increase the likelihood of the adapter settling into a local optimum during its training phase. Based on our experimental data, our approach remains relatively stable against changes in the temperature coefficient. There exists a broad margin within which an optimal-performing adapter can be achieved. In the majority of our experimental tasks, efficient training was realized with temperature coefficients ranging from 1/1000 to 1/10.

## 4 CONCLUSION AND DISCUSSION

Adaptation is already a mature method in computer vision and natural language processing. However, this method has received limited attention in reinforcement learning. We propose an ADAPTER-RL architecture to quickly improve the performance of existing agents in different tasks. This structure can be combined with the intelligence of any adaptation task and some expert methods can be applied. We verified the effectiveness of this method in nanoRTS. This method has a parameter temperature coefficient used to adjust the influence of the base-agent. As our experiments show, choosing an appropriate intermediate value of the temperature coefficient in the range [1/1000, 1/10] usually results in good performance. In reinforcement learning, the strategy of the agent during training often affects exploration and thus the final performance of the agent. A good strategy during training takes into account obtaining more rewards and exploring more state space. The effectiveness of our method depends to a certain extent on the strategy of the base-agent, and the base-agent and adapter complement each other. How to use an adapter to improve a base-agent

with poor strategies may be a future research direction. We expect this approach to be used in more complex practical applications, such as making a base AI specialized for each character it is used in a Multiplayer Online Battle Arena game.

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
