# OpenReview forum: "ADAPTER-RL: Adaptation of Any Agent using Reinforcement Learning"
_ICLR.cc/2024/Conference — ICLR 2024 Conference Withdrawn Submission_

### Official Review · Reviewer_UG4J · 2023-10-23

**Soundness:** 2 fair
**Presentation:** 1 poor
**Contribution:** 1 poor
**Rating:** 3
**Confidence:** 4

**Summary:**

This paper proposes an adapter-based approach to adapt reinforcement learning agents to new tasks. The method uses a separate adapter module that adjusts the outputs of a base agent using proximal policy optimization. Experiments are conducted on RTS games.

**Strengths:**

- The idea of using adapters for transfer in RL could be beneficial for sample efficiency and overcoming catastrophic forgetting.

- Adapters provide a modular approach for incremental learning without interfering with the base agent.

- Method can work with any base agent including rule-based and neural network agents.

**Weaknesses:**

1. The novelty is limited. Adapters have been widely studied for transfer learning in NLP and computer vision. Their application to RL is straightforward with no new techniques proposed.
2. There is no analysis on the learned adapter parameters to provide insights into how adaptation is occurring. Visualizations or other analysis would strengthen the approach.
3. The experimental evaluation is weak. Testing is limited to a simple RTS game with no comparisons to other multi-task or transfer RL techniques. More complex domains should be evaluated.
4. Only win rate and training curves are reported. Standard RL metrics around sample efficiency, reward, etc. should be included. Ablations on adapter design choices are also lacking.
5. The proposed method relies heavily on the base agent capabilities. Performance when base agent is sub-optimal is not explored.
6. The temperature coefficient analysis claims a stable range of [1/1000, 1/10] but there is no clear justification for this range.
There is no evaluation on real-world robotics tasks. The approach may not transfer from simulation.

**Questions:**

1. Have other adapter design choices besides the temperature coefficient been explored?
2. How well does the method transfer to other RL domains beyond RTS games?
3. Is the stable temperature coefficient range task-dependent or generalizable?
4. How does the approach perform when the base agent skills are sub-optimal for the task?

---

### Official Review · Reviewer_dR9v · 2023-10-27

**Soundness:** 2 fair
**Presentation:** 2 fair
**Contribution:** 1 poor
**Rating:** 3
**Confidence:** 4

**Summary:**

This paper studies the problem of adaptation in Reinforcement Learning (RL) and proposes an adaptation strategy to improve training efficiency. This method is compatible with pre-trained neural networks and rule-based agents. The authors provide experiments in the nanoRTS environment to demonstrate the strength of their method.

**Strengths:**

(1) Interesting topic: Adaptation is an interesting and motivating area for reinforcement learning.

**Weaknesses:**

(1) Lack of related works discussion. The authors state that adaptation for RL is a largely unexplored area, while I would recommend doing a further literature review on similar topics. Some examples contain [1, 2, 3]

(2) Insufficient experiments. The provided experiment results are limited. I would recommend doing more experiments with diverse tasks. For example, if the domain is focused on Reinforcement learning, you may try other tasks in the OpenAI gym [4].

(3) Lack of README file to run the code. I appreciate the authors for providing the code. However, it would further benefit the reviewers, and potential users with a README file to guide how to run the code.

Reference:

[1] Nagabandi, Anusha, et al. "Learning to adapt in dynamic, real-world environments through meta-reinforcement learning." arXiv preprint arXiv:1803.11347 (2018).
[2] Arndt, Karol, et al. "Meta reinforcement learning for sim-to-real domain adaptation." 2020 IEEE International Conference on Robotics and Automation (ICRA). IEEE, 2020.
[3] Guez, Arthur, David Silver, and Peter Dayan. "Efficient Bayes-adaptive reinforcement learning using sample-based search." Advances in neural information processing systems 25 (2012).
[4] Brockman, Greg, et al. "Openai gym." arXiv preprint arXiv:1606.01540 (2016).

**Questions:**

(1) What is the relationship between the proposed method with adaptive RL such as [5, 6, 7]

Reference:

[1] Nagabandi, Anusha, et al. "Learning to adapt in dynamic, real-world environments through meta-reinforcement learning." arXiv preprint arXiv:1803.11347 (2018).
[2] Arndt, Karol, et al. "Meta reinforcement learning for sim-to-real domain adaptation." 2020 IEEE International Conference on Robotics and Automation (ICRA). IEEE, 2020.
[3] Guez, Arthur, David Silver, and Peter Dayan. "Efficient Bayes-adaptive reinforcement learning using sample-based search." Advances in neural information processing systems 25 (2012).
[4] Brockman, Greg, et al. "Openai gym." arXiv preprint arXiv:1606.01540 (2016).
[5] Khan, Said G., et al. "Reinforcement learning and optimal adaptive control: An overview and implementation examples." Annual reviews in control 36.1 (2012): 42-59.
[6] Rigter, Marc, Bruno Lacerda, and Nick Hawes. "Risk-averse bayes-adaptive reinforcement learning." Advances in Neural Information Processing Systems 34 (2021): 1142-1154.
[7] Eghbal-zadeh, Hamid, Florian Henkel, and Gerhard Widmer. "Context-adaptive reinforcement learning using unsupervised learning of context variables." NeurIPS 2020 Workshop on Pre-registration in Machine Learning. PMLR, 2021.

---

### Official Review · Reviewer_tvEM · 2023-10-30

**Soundness:** 2 fair
**Presentation:** 2 fair
**Contribution:** 2 fair
**Rating:** 3
**Confidence:** 4

**Summary:**

This paper applies an “adapter” module to RL agents.

**Strengths:**

The method seems reasonable.

**Weaknesses:**

* My main concern with this paper, and why I’m voting for rejection, is that it does not have a related work section that compares other adaptation techniques in RL, and does not compare the method introduced with baselines from prior work. It is hard to gauge the merits if this work without proper discussion of related work (specifically, adaptation methods in RL).
* The text in the plots in Figure 3 is unreadable.
* Typos:
    * Top of 2.3 “Which is a actor-critic paradigm” (PPO is not a paradigm, and should be “an”).
    * Others + clunky wording throughout—please read through the paper with a careful eye.

**Questions:**

* None.